# Prevalence and incidence of diabetic retinopathy in patients with diabetes of Latin America and the Caribbean: A systematic review and meta-analysis

Sebastian A. Medina-Ramirez[1], David R. Soriano-Moreno[1]*, Kimberly G. Tuco[1], Sharong D. Castro-Diaz[1], Rosa Alvarado-Villacorta[2], Josmel Pacheco-Mendoza[3], Marlon Yovera-Aldana[4,5]

1 Unidad de Investigación Clínica y Epidemiológica, Escuela de Medicina, Universidad Peruana Unión, Lima, Peru, 2 Instituto Nacional de Oftalmología, Dr. Francisco Contreras Campos, Lima, Peru, 3 Universidad Científica del Sur, Lima, Peru, 4 Grupo de Investigación de Neurociencias, Efectividad Clínica y Salud Pública, Universidad Científica del Sur, Lima, Peru, 5 Escuela de Posgrado, Universidad Privada Antenor Orrego, Trujillo, Peru

* daresoriano@gmail.com, davidsoriano@upeu.edu.pe

## Abstract

### Objectives

This systematic review aimed to assess the prevalence and incidence of diabetic retinopathy in patients with diabetes of Latin America and the Caribbean.

### Methods

We searched Web of Science (WoS)/Core Collection, WoS/MEDLINE, WoS/Scielo, Scopus, PubMed/Medline and Embase databases until January 16, 2023. We meta-analyzed prevalences according to type 1 diabetes mellitus (T1DM) and type 2 diabetes mellitus (T2DM).

### Results

Forty-three prevalence studies (47 585 participants) and one incidence study (436 participants) were included. The overall prevalence of retinopathy in patients with T1DM was 40.6% (95% CI: 34.7 to 46.6; $I^2$: 92.1%) and in T2DM was 37.3% (95% CI: 31.0 to 43.8; $I^2$: 97.7), but the evidence is very uncertain (very low certainty of evidence). In meta-regression, we found that age (T1DM) and time in diabetes (T2DM) were factors associated with the prevalence. On the other hand, one study found a cumulative incidence of diabetic retinopathy of 39.6% at 9 years of follow-up.

### Conclusions

Two out of five patients with T1DM or T2DM may present diabetic retinopathy in Latin America and the Caribbean, but the evidence is very uncertain. This is a major public health

**Data Availability Statement:** All relevant data are within the manuscript and its Supporting Information files.

**Funding:** The author(s) received no specific funding for this work.

**Competing interests:** The authors have declared that no competing interests exist.

problem, and policies and strategies for early detection and opportunely treatment should be proposed.

## Introduction

Diabetic retinopathy is one of the leading causes of blindness and low vision worldwide and the leading cause of irreversible blindness in adults of productive age [1]. However, the burden of the disease in diabetics varies by geographic area and type of diabetes mellitus (DM), due to differences related to regional prevalence, life expectancy of different populations, as well as social and economic factors [2, 3].

The overall prevalence of diabetic retinopathy is 22.3% and is predicted to increase by a further 50% over the next 25 years [4]. However, there has been a trend towards a lower incidence in the last three decades [5], as a result of the effectiveness of intensive control of DM [6]. In Latin America, health coverage in the general population fluctuates between 45 and 97% [7], and only 2 out of 5 patients with DM have adequate glucose control [8].

A systematic review with meta-analysis was carried out at a global level and found a prevalence of 22.3% [4]. Another review conducted in Asia reported a prevalence of 28% [9]. Likewise, a systematic review reported a prevalence in Africa that varied between 30.2% and 31.6% [10]. Additionally, a systematic review evaluated the annual incidence of diabetic retinopathy in Asia, North America, the Caribbean and Africa varied from 2.2% to 12.7% [11]. However, in our population there is a knowledge gap about the prevalence and incidence of diabetic retinopathy. This is relevant since the prevalence can be influenced by various factors, such as the type of population evaluated, the type of DM, the time of onset of the disease, the different treatments applied, the age of the patients, among others [12–14]. In addition, it is important to highlight that existing reviews do not usually evaluate the prevalence and incidence of diabetic retinopathy according to the type of DM. Therefore, this systematic review aimed to assess the prevalence and incidence of diabetic retinopathy in patients with diabetes of Latin America and the Caribbean.

## Methods

We performed a systematic review following the methodology proposed by JBI Manual for Evidence Synthesis: Systematic reviews of prevalence and incidence [15] and the Preferred Reporting Items for Systematic Reviews and Meta-Analysis (PRISMA) guidelines 2020 [16] (**S1 Table**). The study protocol has been registered at PROSPERO, number CRD42021231181.

### Eligibility criteria

We included cross-sectional and cohort studies that reported the prevalence or incidence of diabetic retinopathy in patients with type 1 DM (T1DM) or type 2 DM (T2DM) in different settings (general population, populations accessing primary care services or hospital populations) and conducted in Latin America and the Caribbean. The definition of DM was taken from what was reported by the study, in case the study did not explicitly mention the type of diabetes, it was considered as T1DM and T2DM. The following classifications were considered to assess the status of diabetic retinopathy: Early Treatment Diabetic Retinopathy Study (ETDRS) classification [17], Proposed International Clinical Scale of Diabetic Retinopathy Global Diabetic Retinopathy Project Group (GDRPG) [18], Scottish Diabetic Retinopathy Grading Scheme (SDRGS) [19], among others. We excluded studies in other types of diabetes

(gestational diabetes, etc.), duplicate populations, clinical trials, case-control studies, case reports, editorials, commentaries, clinical practice guidelines, opinions, reviews, manuscripts not available in full text, and we opted to exclude studies with a sample size of fewer than 50 patients due to the potential impact on the reliability and generalizability of the findings.

## Literature search and study selection

A systematic search was performed in six sources: Web of Science (WoS)/Core Collection, WoS/MEDLINE, WoS/Scielo, Scopus, PubMed and Embase until January 16, 2023. Despite planning to search the Dimensions database in the protocol, the authors were unable to search this database due to a lack of access. There were no restrictions regarding language or date of publication. The complete search strategy for each database is available in the (**S2 Table**). We also reviewed the reference list of all included studies to find additional eligible studies.

References found were exported to the Rayyan program [20] and duplicates were manually removed by one author (SAMR). Subsequently, two investigators (KGT and SDCD) independently screened the articles by titles and abstracts to identify potentially relevant articles for inclusion. The chosen studies went on to full-text review independently by the authors (KGT, SDCD, and SAMR). Discrepancies were resolved by consensus at a meeting with a third author (DRSM).

## Data extraction

The authors (KGT, SDCD, and SAMR) independently extracted the following data of interest using a Microsoft Excel sheet: author, year of publication, country, study design, setting, sample size, age, sex, glycosylated hemoglobin (HbA1c), duration of DM, type of DM, classification of diabetic retinopathy, and prevalence/incidence of retinopathy. Another author (DRSM) resolved the discrepancies. In case of duplicate populations, the most complete study was included.

## Risk of bias

The authors (SAMR, KGT, and SDCD) independently assessed the methodological quality of prevalence and incidence studies using the JBI Critical Appraisal Tool [21]. Another author (DRSM) resolved discrepancies at this stage. This scale has 9 items with a maximum score of 9 points, considering if it meets a criterion 1 point and if it does not meet or is unclear 0 points. The higher the score, the better the methodological quality.

## Statistical analyses

The prevalence of retinopathy according to type of DM was calculated using the number of patients with diabetes as the denominator and the number of cases of diabetic retinopathy as the numerator, enabling us to calculate the prevalence and were meta-analyzed using a random-effects model. Confidence intervals at 95% were obtained using the exact method. To stabilize variances, we used the Freeman-Tukey Double Arcsine transformation [22]. Articles that did not specify the type of DM were not included in the meta-analysis. To assess heterogeneity and its sources, we used the Cochrane Q statistic, the $I^2$ test and performed subgroup analyses according to degree of retinopathy, country, year of study, sex, setting, and diagnostic criteria of retinopathy [4, 23, 24]. In addition, we performed bivariate and multivariate meta-regression analyses to assess factors that might influence the prevalence of retinopathy. As a post-hoc analysis we performed a meta-regression. As covariates we included: age, time on DM, study risk of bias score, and year of publication. HbA1c was not included as a covariate

due to the low number of studies with this data. In the multivariate analysis, variables with a p<0.05 in the bivariate analysis were included along with the age variable. We assessed the effect of small studies by visual inspection of the Funnel plot and Egger's test. We considered a p<0.05 as statistically significant. Analyses were performed with STATA V16.0 software.

### Evidence certainty assessment

We assessed the certainty of the evidence of the prevalence of diabetic retinopathy in Latin America and the Caribbean using the GRADE approach. For this evaluation, we considered the risk of study bias, imprecision (sample sizes and CI), indirect evidence, inconsistency (heterogeneity), and publication bias [25]. We adapted the assessment to prevalence estimates. The certainty of the evidence was characterized as high, moderate, low, or very low. We communicated the findings of the main results using the informative statements proposed by GRADE [26]. Results were reported in a Summary of Findings table (SoF).

## Results

### Search results

We identified 2739 studies after the duplicate elimination process. A total of 256 full-text articles were reviewed. After applying the eligibility criteria, we included 43 prevalence studies [27–69] and one incidence study [70] (Fig 1). The reasons for exclusion of the reviewed full-text studies are given in S3 Table.

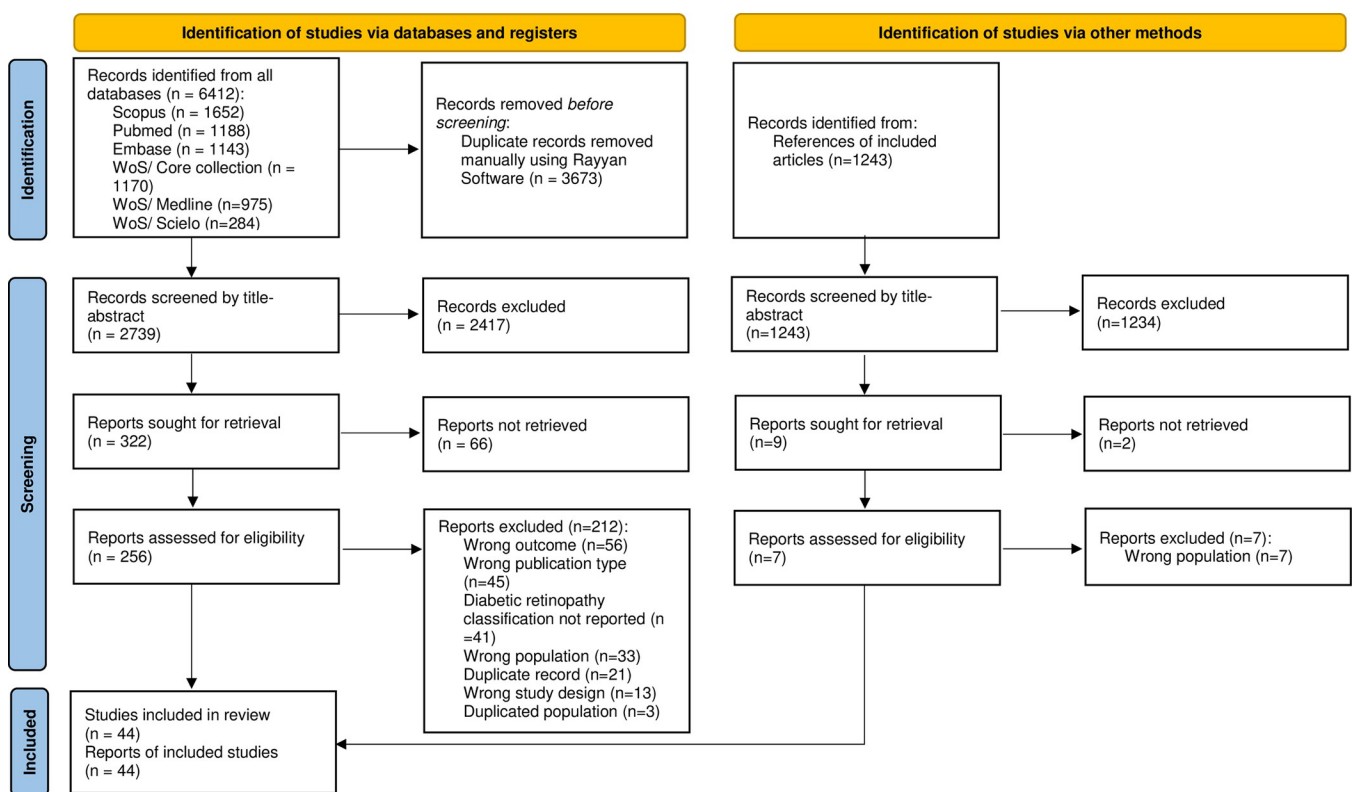

**Fig 1. Flow diagram summarizing the process of literature search and selection.**

## Studies characteristics

The 43 prevalence studies included 47 585 participants and their characteristics are shown in Table 1. With respect to scientific production by country, eighteen studies were conducted in Brazil [30, 32, 33, 35–37, 39–42, 44, 50, 54, 58, 59, 65–67], twelve in Mexico [27–29, 38, 45, 47, 49, 51, 55, 57, 60, 64], four in Cuba [31, 34, 43, 48], two in Peru [46, 61] two in Chile [63, 69], and one in Argentina [68], Costa Rica [62], Ecuador [56], Puerto Rico [52], and Suriname [53].

Regarding the populations assessed, in twenty-four studies the participants were assessed in the hospital [29, 32–35, 37–41, 44–46, 49–52, 56–58, 62, 65, 67, 68], ten in the community [18, 19, 21, 33, 34, 38, 44, 46, 50, 51], and nine in primary care [31, 36, 48, 54, 61, 63, 64, 66, 69]. For the type of DM, eight studies assessed T1DM [32, 34, 35, 39, 40, 44, 50, 58], eighteen T2DM [27, 33, 37, 41, 42, 45, 46, 48, 49, 51, 54–56, 59, 61, 64, 66, 68] and seventeen both types of DM [28–31, 36, 38, 43, 47, 52, 53, 57, 60, 62, 63, 65, 67, 69] of which only one presented prevalences by type of DM. As for the studies in T1DM, they generally assessed young people with a mean age range between 25 and 40 years, including one study in a pediatric population with a mean age of 12 years [32]. On the other hand, studies in patients with T2DM assessed adult patients with a mean age range between 50 to 65 years. In addition, a study in Brazil assessed a population of Xavante Indians [59].

Regarding the diagnostic criteria for diabetic retinopathy, seventeen studies used the ETDRS [27, 30, 32, 33, 35–38, 42, 43, 51–53, 56, 63, 65, 67], thirteen used the GDRPG [39–41, 44–46, 50, 54, 58, 61, 66, 68, 69], six the SDRGS [47, 53, 57, 60, 62, 64], three used the American Academy of Ophthalmology guidelines [28, 29, 49], three used the L' Esperance classification [31, 34, 48], and one used the Revised English Diabetic Eye Screening Program Grading System (REDESPGS) [55].

Only one study reported the incidence of diabetic retinopathy. It was conducted on the island of Barbados and included 436 patients from the community of African origin. They assessed both types of DM, and used the ETDRS as diagnostic criteria [70] (**S4 Table**).

## Diabetic retinopathy prevalence in T1DM and T2DM

Meta-analyses were performed to estimate the prevalence of diabetic retinopathy in Latin American and Caribbean patients according to the type of DM. For T1DM, 9 studies were included with a total of 4505 participants and for T2DM, 19 studies were included with a total of 11 569 participants. In patients with T1DM, the prevalence of diabetic retinopathy was 40.6% (95% CI: 34.7 to 46.6; $I^2$: 92.1%), but the evidence is very uncertain. For those with T2DM, the prevalence was 37.3% (95% CI: 31.0 to 43.8; $I^2$: 97.7%), but the evidence is very uncertain (Table 2 and Fig 2).

## Diabetic retinopathy incidence

The Leske-2006 study found a cumulative incidence of diabetic retinopathy of 39.6% at 9 years of follow-up. In addition, 8.2% of patients with non-proliferative retinopathy (NPDR) at the start of follow-up progressed to proliferative retinopathy (PDR). The incidences between age and sex were similar (S4 Table).

## Subgroup analyses in T1DM

We assessed the prevalence of diabetic retinopathy according to degree of retinopathy, country, year, sex, setting, and diagnostic criteria. In patients with T1DM, the prevalence of NPDR and PDR was 25.0% (95% CI:19.8 to 30.5; $I^2$: 92.1%) and 12.4% (95% CI: 8.3 to 17.2; $I^2$: 93.4%), respectively.

**Table 1. Characteristics of the included studies assessing the prevalence of diabetic retinopathy in Latin America and the Caribbean (n = 40).**

| Author—year | Country | Setting | Sample size | DM type | Age (mean ± SD years), male (%), diabetes time (mean ± SD years), A1c (mean ± SD %) | Diabetic retinopathy diagnostic method | Diabetic retinopathy prevalence | | | Quality score (Max. 9) |
|---|---|---|---|---|---|---|---|---|---|---|
| | | | | | | | Total | NPDR | PDR | |
| Arenas-Cavalli–2022 [69] | Chile | Primary care | 1123 | T1DM/T2DM | Age: 63.0 ± 12.7 | GDRPG | 21.3% | General: 11.2% | 0.4% | 6 |
| | | | | | Male: 40.2% | | | Mild: 1.9% | | |
| | | | | | Diabetes time: NR | | | Moderate: 7.2% | | |
| | | | | | HbA1c: NR | | | Severe: 2.1% | | |
| Ortiz-Basso–2022 [68] | Argentina | Hospital | 2743 | T2DM | Age: 60.1 ± 12.0 | GDRPG | 21.6% | General: 21.3% | 0.3% | 7 |
| | | | | | Male: 41.5% | | | Mild: 11.3% | | |
| | | | | | Diabetes time: NR | | | Moderate: 7.2% | | |
| | | | | | HbA1c: 7.2 ± 1.8 | | | Severe: 2.7% | | |
| Galvão—2021 [67] | Brazil | Hospital | 219 | T1DM/T2DM | Age: NR | ETDRS | 32.0% | General: 24.2% | 7.8% | 5 |
| | | | | | Male: 40.2% | | | Mild: NR | | |
| | | | | | Diabetes time: NR | | | Moderate: NR | | |
| | | | | | HbA1c: NR | | | Severe: NR | | |
| Graue-Hernandez–2020 [64] | Mexico | Primary care | 1232 | T2DM | Age: 51.5 ± 10.0 | SDRGS | 17.3% | General: 17.0% | 0.3% | 7 |
| | | | | | Male: 44.2% | | | Mild: 13.8% | | |
| | | | | | Diabetes time: 2.0 ± 3.7 | | | Moderate: 2.4% | | |
| | | | | | HbA1c: 8.1 ± 2.8 | | | Severe: 0.7% | | |
| Abalem—2020 [65] | Brazil | Hospital | 234 | T1DM/T2DM | Age: 59.6 ± 13.0 | ETDRS | 65.0% | General: 54.7% | 10.3% | 5 |
| | | | | | Male: 50.4% | | | Mild: 18.4% | | |
| | | | | | Diabetes time: 15.9 ± 8.8 | | | Moderate: 19.7% | | |
| | | | | | HbA1c: 8.1 ± 1.8 | | | Severe: 16.7% | | |
| Ben—2020 [66] | Brazil | Primary care | 206 | T2DM | Age: 63.5 ± 10.6 | GDRPG | 23.8% | General: NR | NR | 5 |
| | | | | | Male: 39.3% | | | Mild: NR | | |
| | | | | | Diabetes time: 6.9 ± 7.8 | | | Moderate: NR | | |
| | | | | | HbA1c: 7.8 ± 1.9 | | | Severe: NR | | |
| Adrianzén–2019 [61] | Peru | Primary care | 3239 | T2DM | Age: 59.0 ± 11.7 | GDRPG | 25.9% | General: 23.1% | 2.7% | 7 |
| | | | | | Male: 37.3% | | | Mild: 14.1% | | |
| | | | | | Diabetes time: NR | | | Moderate: 6.4% | | |
| | | | | | HbA1c: NR | | | Severe: 2.6% | | |

(*Continued*)

**Table 1.** (Continued)

| Author—year | Country | Setting | Sample size | DM type | Age (mean ± SD years), male (%), diabetes time (mean ± SD years), A1c (mean ± SD %) | Diabetic retinopathy diagnostic method | Diabetic retinopathy prevalence | | | Quality score (Max. 9) |
|---|---|---|---|---|---|---|---|---|---|---|
| | | | | | | | Total | NPDR | PDR | |
| Acevedo– 2019 [62] | Costa Rica | Hospital | 553 | T1DM/ T2DM | Age: NR | SDRGS | 20.1% | General: 18.4% | 1.6% | 6 |
| | | | | | Male: NR | | | Mild: 13.0% | | |
| | | | | | Diabetes time: NR | | | Moderate: 2.2% | | |
| | | | | | HbA1c: NR | | | Severe: 3.3% | | |
| Avendaño-Veloso– 2019 [63] | Chile | Primary care | 6784 | T1DM/ T2DM | Age: 61.0 ± 9.8 | ETDRS | 14.9% | General: 14.2% | 0.7% | 5 |
| | | | | | Male: 34.6% | | | Mild: 7.2% | | |
| | | | | | Diabetes time: NR | | | Moderate: 4.6% | | |
| | | | | | HbA1c: NR | | | Severe: 2.4% | | |
| Lopez-Ramos– 2018 [57] | Mexico | Hospital | 1565 | T1DM/ T2DM | Age: NR | SDRGS | 15.6% | General: 11.3% | 4.3% | 8 |
| | | | | | Male: 34.6% | | | Mild: 7.6% | | |
| | | | | | Diabetes time: NR | | | Moderate: 2% | | |
| | | | | | HbA1c: NR | | | Severe: 1.7% | | |
| Nunes Melo– 2018 [58] | Brazil | Hospital | 1644 | T1DM | Age: 30.1 ± 12.0 | GDRPG | 35.8% | General: 25.4% | 10.5% | 6 |
| | | | | | Male: 44.2% | | | Mild: 18.1% | | |
| | | | | | Diabetes time: 15.3 ± 9.3 | | | Moderate: 6.6% | | |
| | | | | | HbA1c: 9.0 ± 2.1 | | | Severe: 0.7% | | |
| Lima– 2018 [59] | Brazil | Community | 140 | T2DM | Age: 52.9 | ETDRS | 19.3% | General: 18.6% | 0.7% | 4 |
| | | | | | Male: 28.6% | | | Mild: 6.4% | | |
| | | | | | Diabetes time: NR | | | Moderate: 6.4% | | |
| | | | | | HbA1c: NR | | | Severe: 5.7% | | |
| Lopez-Star– 2018 [60] | Mexico | Community | 562 | T1DM/ T2DM | Age: NR | SDRGS | 44.7% | General: 39.1% | 5.5% | 7 |
| | | | | | Male: NR | | | Mild: 26.7% | | |
| | | | | | Diabetes time: NR | | | Moderate: 8.5% | | |
| | | | | | HbA1c: NR | | | Severe: 2.3% | | |
| Rosses– 2017 [54] | Brazil | Primary care | 219 | T2DM | Age: 64.9 ± 11.0 | GDRPG | 25.1% | General: 16.4% | 1.4% | 5 |
| | | | | | Male: 40.2% | | | Mild: 3.2% | | |
| | | | | | Diabetes time: 7.6 ± 8.2 | | | Moderate: 11.0% | | |
| | | | | | HbA1c: 7.2 ± 1.7 | | | Severe: 2.3% | | |

(*Continued*)

**Table 1.** (Continued)

| Author—year | Country | Setting | Sample size | DM type | Age (mean ± SD years), male (%), diabetes time (mean ± SD years), A1c (mean ± SD %) | Diabetic retinopathy diagnostic method | Diabetic retinopathy prevalence | | | Quality score (Max. 9) |
|---|---|---|---|---|---|---|---|---|---|---|
| | | | | | | | Total | NPDR | PDR | |
| Mendoza-Herrera– 2017 [55] | Mexico | Community | 1000 | T2DM | Age: 57.2 ± 11.0 | REDESPGS | 31.7% | General: 24.6% | 7.1% | 5 |
| | | | | | Male: 27% | | | Mild: NR | | |
| | | | | | Diabetes time: NR | | | Moderate: NR | | |
| | | | | | HbA1c: NR | | | Severe: NR | | |
| Flores-Mena– 2017 [56] | Ecuador | Hospital | 88 | T2DM | Age: 48.9 ± 9.6 | ETDRS | 68.2% | General: 64.8% | 3.4% | 5 |
| | | | | | Male: NR | | | Mild: NR | | |
| | | | | | Diabetes time: NR | | | Moderate: NR | | |
| | | | | | HbA1c: NR | | | Severe: NR | | |
| Rodriguez– 2016 [52] | Puerto Rico | Hospital | 411 | T1DM/ T2DM | Age: 56.1 ± 9.9 | ETDRS | 40.1% | General: 35.0% | 5.1% | 7 |
| | | | | | Male: 29.9% | | | Mild: 22.6% | | |
| | | | | | Diabetes time: 12.5 ± 8.0 | | | Moderate: 7.8% | | |
| | | | | | HbA1c: NR | | | Severe: 4.6% | | |
| Minderhoud– 2016 [53] | Suriname | Community | 689 | T1DM/ T2DM | Age: NR | SDRGS | 19.0% | General: 15.2% | 3.8% | 6 |
| | | | | | Male: 39.8% | | | Mild: 7.8% | | |
| | | | | | Diabetes time: 12.3 ± 11.1 | | | Moderate: 2.9% | | |
| | | | | | HbA1c: NR | | | Severe: 4.5% | | |
| Malerbi– 2015 [50] | Brazil | Hospital | 1266 | T1DM | Age: 31.0 ± 12.0 | GDRPG | 47.2% | General: 34.6% | 12.6% | 6 |
| | | | | | Male: 43.2% | | | Mild: 27.8% | | |
| | | | | | Diabetes time: 16.0 ± 9.0 | | | Moderate: 4.1% | | |
| | | | | | HbA1c: NR | | | Severe: 2.7% | | |
| Cepeda-Nieto– 2015 [51] | Mexico | Hospital | 177 | T2DM | Age: 59.6 ± 10.2 | ETDRS | 68.4% | General: 28.2% | 40.1% | 4 |
| | | | | | Male: 61% | | | Mild: NR | | |
| | | | | | Diabetes time: NR | | | Moderate: NR | | |
| | | | | | HbA1c: NR | | | Severe: NR | | |
| Valdés—2013 [48] | Cuba | Primary care | 150 | T2DM | Age: 49.2 ± 9.5 | L' Esperance | 6.0% | General: NR | NR | 4 |
| | | | | | Male: 44% | | | Mild: NR | | |
| | | | | | Diabetes time: NR | | | Moderate: NR | | |
| | | | | | HbA1c: NR | | | Severe: NR | | |

(*Continued*)

**Table 1.** (Continued)

| Author—year | Country | Setting | Sample size | DM type | Age (mean ± SD years), male (%), diabetes time (mean ± SD years), A1c (mean ± SD %) | Diabetic retinopathy diagnostic method | Diabetic retinopathy prevalence | | | Quality score (Max. 9) |
|---|---|---|---|---|---|---|---|---|---|---|
| | | | | | | | Total | NPDR | PDR | |
| Alcaraz– 2013 [49] | Mexico | Hospital | 100 | T2DM | Age: 55.5 ± 11.7 | AAO guidelines | 48.0% | General: 42.0% | 6.0% | 7 |
| | | | | | Male: 47% | | | Mild: 23.0% | | |
| | | | | | Diabetes time: 8.7 ± 7.7 | | | Moderate: 14.0% | | |
| | | | | | HbA1c: NR | | | Severe: 5.0% | | |
| Polack– 2012 [47] | Mexico | Community | 335 | T1DM/T2DM | Age: NR | SDRGS | 40.3% | General: 31.3% | 9.0% | 6 |
| | | | | | Male: NR | | | Mild: 16.7% | | |
| | | | | | Diabetes time: NR | | | Moderate: 8.7% | | |
| | | | | | HbA1c: NR | | | Severe: 6.0% | | |
| Perera—2011 [43] | Cuba | Community | 150 | T1DM/T2DM | Age: NR | ETDRS | 16.0% | General: NR | NR | 5 |
| | | | | | Male: 26% | | | Mild: NR | | |
| | | | | | Diabetes time: NR | | | Moderate: NR | | |
| | | | | | HbA1c: NR | | | Severe: NR | | |
| Almeida– 2011 [44] | Brazil | Hospital | 150 | T1DM | Age: 41.3 ± 8.6 | GDRPG | 56.7% | General: NR | NR | 3 |
| | | | | | Male: NR | | | Mild: NR | | |
| | | | | | Diabetes time: 31.8 ± 9.2 | | | Moderate: NR | | |
| | | | | | HbA1c: 8.4 ± 1.7 | | | Severe: NR | | |
| Carrillo-Alarcón– 2011 [45] | Mexico | Hospital | 117 | T2DM | Age: 58.1 ± 11.1 | GDRPG | 33.3% | General: 29.9% | 3.4% | 6 |
| | | | | | Male: 22.2% | | | Mild: 21.4% | | |
| | | | | | Diabetes time: 9.9 ± 6.4 | | | Moderate: 6.0% | | |
| | | | | | HbA1c: NR | | | Severe: 2.6% | | |
| Villena– 2011 [46] | Peru | Hospital | 1222 | T2DM | Age: 59.3 ± 5.9 | GDRPG | 23.1% | General: 20.4% | 2.7% | 5 |
| | | | | | Male: NR | | | Mild: 10.2% | | |
| | | | | | Diabetes time: 5.6 ± 7.4 | | | Moderate: 8.5% | | |
| | | | | | HbA1c: NR | | | Severe: 1.6% | | |
| Rodrigues– 2010 [40] | Brazil | Hospital | 573 | T1DM | Age: 33.0 ± 13.0 | GDRPG | 44.7% | General: 22.0% | 22.7% | 7 |
| | | | | | Male: 50.4% | | | Mild: 15.0% | | |
| | | | | | Diabetes time: 16.0 ± 9.0 | | | Moderate: 4.0% | | |
| | | | | | HbA1c: NR | | | Severe: 3.0% | | |

(*Continued*)

**Table 1.** (Continued)

| Author—year | Country | Setting | Sample size | DM type | Age (mean ± SD years), male (%), diabetes time (mean ± SD years), A1c (mean ± SD %) | Diabetic retinopathy diagnostic method | Diabetic retinopathy prevalence | | | Quality score (Max. 9) |
|---|---|---|---|---|---|---|---|---|---|---|
| | | | | | | | Total | NPDR | PDR | |
| Preti– 2010 [41] | Brazil | Hospital | 105 | T2DM | Age: 63.5 ± 10.0 | GDRPG | 85.7% | General: 50.5% | 35.2% | 6 |
| | | | | | Male: 45.7% | | | Mild: 28.6% | | |
| | | | | | Diabetes time: NR | | | Moderate: 17.1% | | |
| | | | | | HbA1c: NR | | | Severe: 4.8% | | |
| Sawitzki– 2010 [42] | Brazil | Community | 120 | T2DM | Age: 63.5 | ETDRS | 38.3% | General: 34.2% | 4.2% | 7 |
| | | | | | Male: 48.3% | | | Mild: NR | | |
| | | | | | Diabetes time: NR | | | Moderate: NR | | |
| | | | | | HbA1c: NR | | | Severe: NR | | |
| Prado-Serrano– 2009 [38] | Mexico | Hospital | 13670 | T1DM/ T2DM | Age: 65.5 | ETDRS | 71.0% | General: 26.3% | 44.7% | 5 |
| | | | | | Male: 39% | | | Mild: NR | | |
| | | | | | Diabetes time: NR | | | Moderate: NR | | |
| | | | | | HbA1c: NR | | | Severe: NR | | |
| Esteves– 2009 [39] | Brazil | Hospital | 437 | T1DM | Age: 26.8 ± 7.8 | GDRPG | 44.4% | General: 22.2% | 22.2% | 7 |
| | | | | | Male: 50.3% | | | Mild: 15.1% | | |
| | | | | | Diabetes time: 14.4 ± 7.3 | | | Moderate: 4.1% | | |
| | | | | | HbA1c: NR | | | Severe: 3.0% | | |
| Gonçalves– 2008 [36] | Brazil | Primary care | 2223 | T1DM/ T2DM | Age: 59.3 ± 12.0 | ETDRS | 25.5% | General: 22.1% | 3.3% | 5 |
| | | | | | Male: 29.5% | | | Mild: NR | | |
| | | | | | Diabetes time: 8.1 ± 6.3 | | | Moderate: NR | | |
| | | | | | HbA1c: NR | | | Severe: NR | | |
| Lisboa– 2008 [37] | Brazil | Hospital | 90 | T2DM | Age: 57.3 ± 9.4 | ETDRS | 34.4% | General: 23.3% | 11.1% | 6 |
| | | | | | Male: 44.4% | | | Mild: NR | | |
| | | | | | Diabetes time: 13.3 ± 7.0 | | | Moderate: NR | | |
| | | | | | HbA1c: 9.1 ± 1.6 | | | Severe: NR | | |
| Sampaio– 2007 [35] | Brazil | Hospital | 81 | T1DM | Age: 26.4 ± 8.7 | ETDRS | 21.0% | General: 8.6% | 12.3% | 5 |
| | | | | | Male: 35.8% | | | Mild: NR | | |
| | | | | | Diabetes time: 13.4 ± 5.8 | | | Moderate: NR | | |
| | | | | | HbA1c: 10.1 ± 1.8 | | | Severe: NR | | |
| Licea—2006 [34] | Cuba | Hospital | 240 | T1DM | Age: 30.9 ± 8.0 | L' Esperance | 40.4% | General: 35.4% | 5.0% | 5 |
| | | | | | Male: 44.2% | | | Mild: NR | | |
| | | | | | Diabetes time: 11.8 ± 7.1 | | | Moderate: NR | | |
| | | | | | HbA1c: 7.7 ± 4.7 | | | Severe: NR | | |

*(Continued)*

**Table 1.** (Continued)

| Author—year | Country | Setting | Sample size | DM type | Age (mean ± SD years), male (%), diabetes time (mean ± SD years), A1c (mean ± SD %) | Diabetic retinopathy diagnostic method | Diabetic retinopathy prevalence | | | Quality score (Max. 9) |
|---|---|---|---|---|---|---|---|---|---|---|
| | | | | | | | Total | NPDR | PDR | |
| Santos– 2005 [33] | Brazil | Hospital | 210 | T2DM | Age: 58.7 ± 12.0 | ETDRS | 47.1% | General: NR | NR | 6 |
| | | | | | Male: 32.4% | | | Mild: NR | | |
| | | | | | Diabetes time: 10.5 ± 9.7 | | | Moderate: NR | | |
| | | | | | HbA1c: NR | | | Severe: NR | | |
| Crespo—2004 [31] | Cuba | Primary care | 559 | T1DM/ T2DM | Age: NR | L' Esperance | 20.6% | General: 16.1% | 4.5% | 4 |
| | | | | | Male: 32.7% | | | Mild: NR | | |
| | | | | | Diabetes time: NR | | | Moderate: NR | | |
| | | | | | HbA1c: NR | | | Severe: NR | | |
| Pereira– 2004 [32] | Brazil | Hospital | 81 | T1DM | Age: 12.0 ± 12.0 | ETDRS | 17.3% | General: 14.8% | 2.5% | 3 |
| | | | | | Male: 49.4% | | | Mild: NR | | |
| | | | | | Diabetes time: 5.8 ± 4.4 | | | Moderate: NR | | |
| | | | | | HbA1c: NR | | | Severe: NR | | |
| Alvarenga– 2003 [30] | Brazil | Community | 575 | T1DM/ T2DM | Age: 60.4 ± 9.6 | ETDRS | 51.0% | General: 32.2% | 18.8% | 6 |
| | | | | | Male: 47.5% | | | Mild: 15.7% | | |
| | | | | | Diabetes time: NR | | | Moderate: 12.3% | | |
| | | | | | HbA1c: NR | | | Severe: 4.2% | | |
| Lima-Gómez– 2001 [28] | Mexico | Community | 1472 | T1DM/ T2DM | Age: NR | AAO guidelines | 18.6% | General: 17.1% | 1.6% | 5 |
| | | | | | Male: NR | | | Mild: NR | | |
| | | | | | Diabetes time: NR | | | Moderate: NR | | |
| | | | | | HbA1c: NR | | | Severe: NR | | |
| Lima-Gómez– 2001 [29] | Mexico | Hospital | 621 | T1DM/ T2DM | Age: NR | AAO guidelines | 35.3% | General: NR | NR | 5 |
| | | | | | Male: NR | | | Mild: NR | | |
| | | | | | Diabetes time: NR | | | Moderate: NR | | |
| | | | | | HbA1c: NR | | | Severe: NR | | |
| Villalpando– 1997 [27] | Mexico | Community | 210 | T2DM | Age: 50.4 ± 2.3 | ETDRS | 49.5% | General: 43.8% | 5.7% | 7 |
| | | | | | Male: 39.5% | | | Mild: NR | | |
| | | | | | Diabetes time: NR | | | Moderate: NR | | |
| | | | | | HbA1c: NR | | | Severe: NR | | |

NR: Not reported, DM: Diabetes Mellitus, T1DM: Type 1 Diabetes Mellitus, T2DM: Type 2 Diabetes Mellitus, NDPR: Non-proliferative diabetic retinopathy, PDR: Proliferative diabetic retinopathy, SDRGS: Scottish Diabetic Retinopathy Grading Scheme, ETDRS: Early Treatment Diabetic Retinopathy Study, GDRPG: Global Diabetic Retinopathy Project Group, REDESPGS: Revised English Diabetic Eye Screening Program Grading System, AAO: American Academy of Ophthalmology

**Table 2. Prevalence of diabetic retinopathy and subgroups in Latin America and the Caribbean countries.**

| | | Diabetic retinopathy in T1DM | | | | Diabetic retinopathy in T2DM | | | |
|---|---|---|---|---|---|---|---|---|---|
| | | n studies | Prevalence (%) | 95% CI | $I^2$ (%) | n studies | Prevalence (%) | 95% CI | $I^2$ (%) |
| Overall | | 9 | 40.6 | 34.7 to 46.6 | 92.1 | 19 | 37.3 | 31.0 to 43.8 | 97.7 |
| Retinopathy grade | | | | | | | | | |
| | Non proliferative | 8 | 25.0 | 19.8 to 30.5 | 92.1 | 16 | 30.6 | 26.2 to 35.2 | 95.3 |
| | Mild non proliferative | 5 | 18.7 | 13.5 to 24.5 | 93.7 | 10 | 13.4 | 10.9 to 16.2 | 90.0 |
| | Moderate non proliferative | 5 | 4.8 | 3.3 to 6.5 | 75.0 | 10 | 8.8 | 6.4 to 11.4 | 92.5 |
| | Severe non proliferative | 5 | 2.4 | 0.9 to 4.4 | 89.0 | 10 | 3.5 | 2.1 to 5.1 | 90.9 |
| | Proliferative | 8 | 12.4 | 8.3 to 17.2 | 93.4 | 16 | 5.8 | 3.2 to 9.1 | 97.3 |
| Country | | | | | | | | | |
| | Brazil | 8 | 40.6 | 34.1 to 47.3 | 93.1 | 8 | 42.2 | 27.0 to 58.2 | 97.0 |
| | Mexico | | | | | 6 | 40.6 | 26.3 to 55.8 | 98.1 |
| | Cuba | 1 | 40.4 | 34.2 to 46.9 | | 1 | 6.0 | 2.8 to 11.1 | |
| | Peru | | | | | 2 | 25.1 | 23.8 to 26.4 | |
| | Argentina | | | | | 1 | 21.6 | 20.1 to 23.2 | |
| | Ecuador | | | | | 1 | 68.2 | 57.4 to 77.7 | |
| Year | | | | | | | | | |
| | 1997 to 2010 | 5 | 34.2 | 25.7 to 43.3 | 90.6 | 5 | 51.7 | 35.3 to 68.0 | 95.1 |
| | 2011 to 2022 | 4 | 48.7 | 38.9 to 58.5 | 94.9 | 14 | 32.4 | 26.4 to 38.6 | 97.5 |
| Sex | | | | | | | | | |
| | Male | 5 | 32.3 | 24.2 to 41.0 | 81.5 | 12 | 34.6 | 28.4 to 41.1 | 91.8 |
| | Female | 5 | 34.9 | 27.6 to 42.6 | 79.7 | 12 | 28.5 | 23.2 to 34.2 | 94.0 |
| Setting | | | | | | | | | |
| | Community | | | | | 4 | 34.3 | 23.9 to 45.6 | 92.4 |
| | Primary care | 1 | 63.6 | 45.1 to 79.6 | | 6 | 25.6 | 16.5 to 36.1 | 97.8 |
| | Hospital | 8 | 39.1 | 33.2 to 45.1 | 92.7 | 9 | 47.5 | 34.5 to 60.7 | 98.3 |
| Diagnostic criteria | | | | | | | | | |
| | ETDRS | 3 | 31.8 | 11.1 to 56.9 | | 8 | 48.7 | 36.5 to 60.9 | 94.7 |
| | GDRPG | 5 | 45.1 | 39.0 to 51.4 | 92.8 | 7 | 32.6 | 25.7 to 39.8 | 97.1 |
| | REDESPGS | | | | | 1 | 31.7 | 28.8 to 34.7 | |
| | L' Esperance | 1 | 40.4 | 34.2 to 46.9 | | 1 | 6.0 | 2.8 to 11.1 | |
| | AAO | | | | | 1 | 48.0 | 37.9 to 58.2 | |
| | SDRGS | | | | | 1 | 17.3 | 15.2 to 19.5 | |

T1DM: Type 1 diabetes mellitus, T2DM: type 2 diabetes mellitus, ETDRS: Early Treatment Diabetic Retinopathy Study, GDRPG: Global Diabetic Retinopathy Project Group, REDESPGS: Revised English Diabetic Eye Screening Program Grading System, AAO: American Academy of Ophthalmology, SDRGS: Scottish Diabetic Retinopathy Grading Scheme

On the other hand, when divided by decade of publication, an increase in prevalence was observed. Regarding the setting, a higher frequency is observed in primary centers than in hospitals. Prevalence was very similar for sex, country and diagnostic criteria. The prevalence according to country was made only with data from Brazil and Cuba (Table 2).

## Subgroup analyses in T2DM

In patients with T2DM, the prevalence of NPDR and PDR was 30.6% (95% CI: 26.2 to 35.2; $I^2$: 95.3%) and 5.8% (95% CI: 3.2 to 9.1; $I^2$: 97.3%), respectively. The country with the lowest prevalence was Cuba. The highest prevalence was in Ecuador, close to 70%. With respect to the diagnostic criteria, the studies that used the L'Esperance and SDRGS classifications obtained

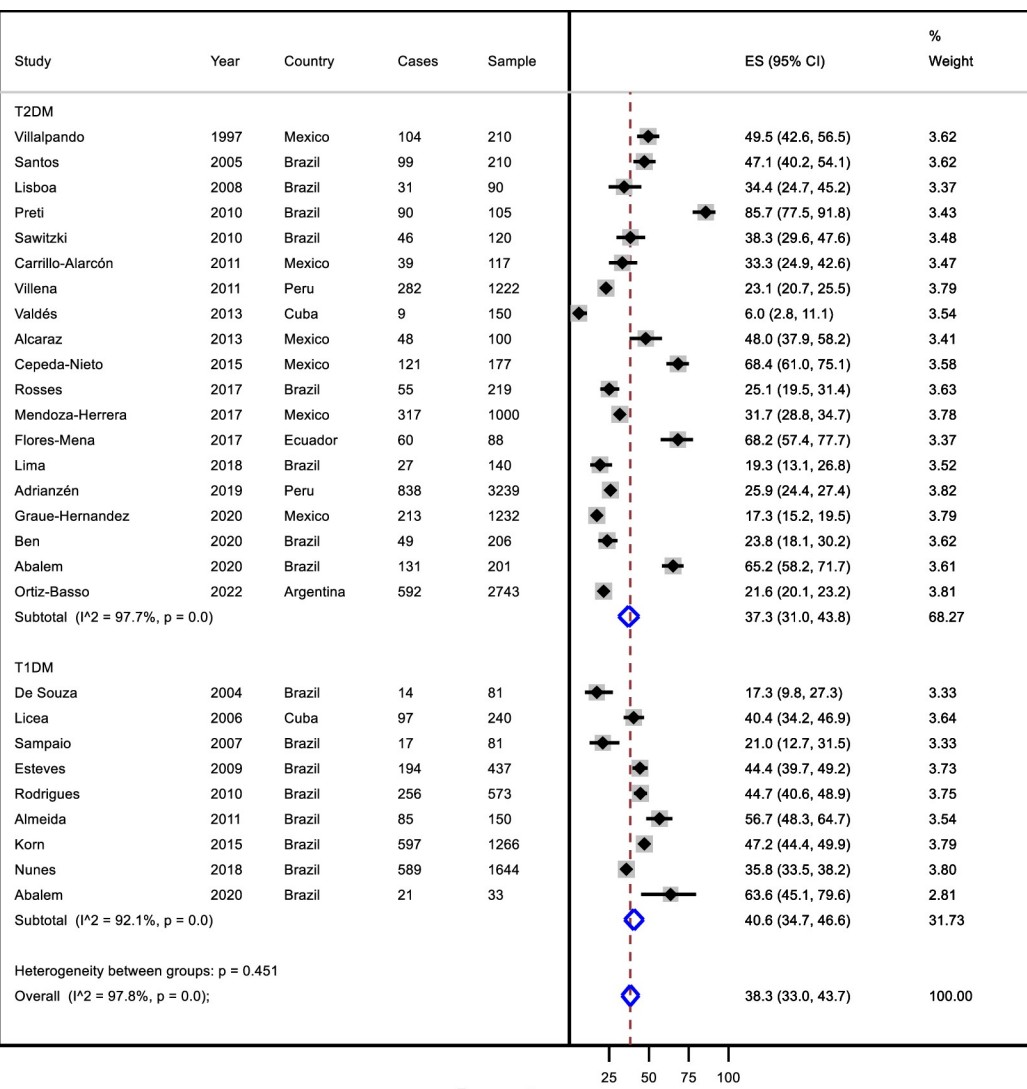

**Fig 2. Prevalence of diabetic retinopathy in Latin American and the Caribbean countries by type of diabetes.**

prevalences lower than 20%. In the other classifications, the prevalence ranges from 30 to 50%. According to the year of publication, there has been a decrease in prevalence in the last decade of almost 20%. There is a higher prevalence in hospital centers than in the general population or primary centers. Finally, similar prevalences were observed according to sex. All the subgroups assessed showed high heterogeneity ($I^2 \geq 70\%$) (Table 2).

DR: diabetic retinopathy. T1DM: Type 1 diabetes mellitus. T2DM: type 2 diabetes mellitus. N: number of studies. P: prevalence. S: sample size.

## Sensitivity analysis

When excluding each individual study, the pooled prevalence of diabetic retinopathy for T1DM ranged from 40.3% (95% CI: 34.2 to 46.4) to 43.0% (95% CI: 36.6 to 49.5) (S1 Fig) and for T2DM ranged from 34.4% (95% CI: 28.4 to 40.6) to 38.4% (95% CI: 31.8 to 45.0) (S2 Fig).

## Meta-regression

In the bivariate meta-regression, the factors explaining the heterogeneity of diabetic retinopathy prevalence were the mean age in patients with T1DM ($\beta$ = 0.013; 95% CI: 0.002 to 0.025; p = 0.031), and time of diabetes in patients with T2DM ($\beta$ = 0.024; 95% CI: 0.009 to 0.039; p = 0.008). In the adjusted analysis, the association of diabetes time in patients with T2DM was maintained ($\beta$ = 0.032; 95% CI: 0.011 to 0.053; p = 0.010) (S5 Table).

## Small study bias (Publication bias)

In the T1DM population, visual inspection of the funnel plot showed a symmetrical distribution for the prevalences of diabetic retinopathy with a nonsignificant Egger's test (p = 1.000) (S3 Fig). On the other hand, for the T2DM population, an asymmetry with bias of small studies with large prevalences was observed (S4 Fig). This finding was corroborated with Egger's test (p = 0.006).

## Risk of bias

The median risk of bias score was 6 [IQR: 5 to 7]. Most of the studies did not present an adequate sampling frame, as they were performed in hospitals or health centers where the prevalence could be overestimated compared to the general population. However, all studies measured diabetic retinopathy with standard criteria. The assessment is summarized in the Fig 3 and detailed in S6 Table.

## Evidence certainty

We assessed the certainty of the evidence for the prevalence of diabetic retinopathy in patients with T1DM and T2DM in Latin America and the Caribbean. For both types of DM, we started

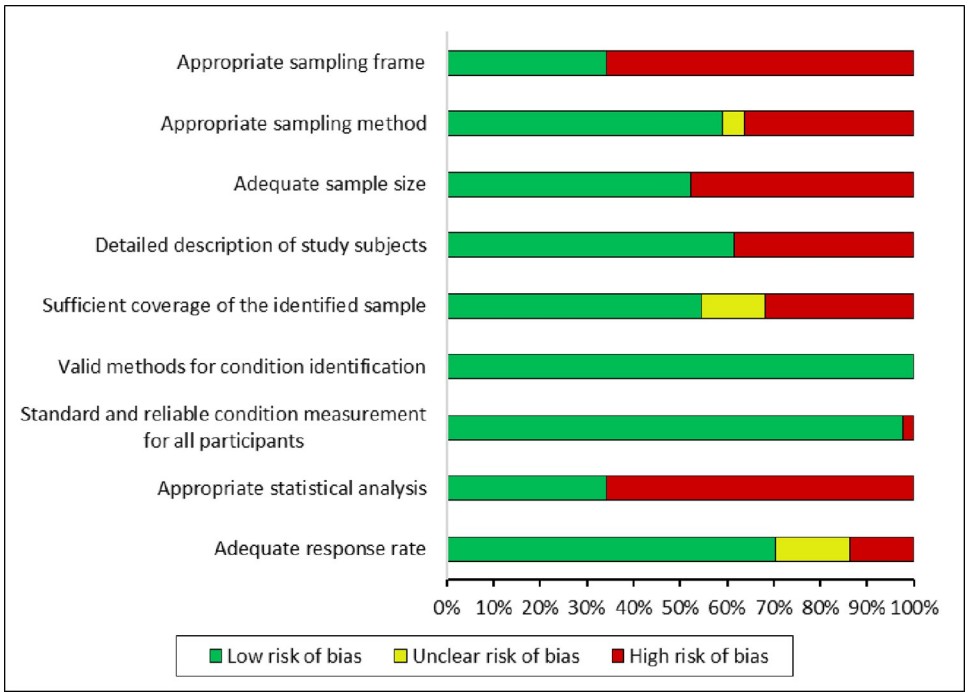

**Fig 3. Results of risk of bias assessment.**

**Table 3. Summary of findings of the prevalence of diabetic retinopathy in Latin America and the Caribbean.**

| Outcomes | Prevalence (95% CI) | | № of participants (Studies) | The certainty of the evidence |
|---|---|---|---|---|
| | Pooled prevalence (%) | 95% CI | | (GRADE) |
| Prevalence of diabetic retinopathy in T1DM in LAC | 40.6 | 34.7 to 46.6 | 4505 patients (9 studies) | ⊕○○○ VERY LOW [a, b, c] |
| Prevalence of diabetic retinopathy in T2DM in LAC | 37.3 | 31.0 to 43.8 | 11 569 patients (19 studies) | ⊕○○○ VERY LOW [d, e, f, g] |

95% CI: Confidence interval at 95%. LAC: Latin America and the Caribbean. T1DM: Type 1 diabetes mellitus. T2DM: type 2 diabetes mellitus.

[a] The certainty rating started from low certainty since 0 population-based studies were included.

[b] High risk of bias, 77% of the included studies had <7 points in the Joana Briggs's scale.

[c] High inconsistency with $I^2$ >70%.

[d] The certainty rating started from low certainty since only 4 population-based studies were included.

[e] High risk of bias, 68% of the included studies had <7 points in the Joana Briggs's scale.

[f] High inconsistency with $I^2$ >70%.

[g] Publication bias was detected in the funnel plot and Egger's test.

with low certainty due to the scarcity of community studies. In addition, we decreased certainty by two levels due to the large number of studies with high or moderate risk of bias according to the JBI scale. Heterogeneity in the meta-analyses was high ($I^2 > 70\%$). Finally, for the T2DM population we decreased certainty by publication bias present in the funnel plot and Egger's test (Table 3). We did not assess the certainty of the evidence of incidence because it was only one article and did not differentiate between both types of DM.

## Discussion

### Main findings

In this systematic review and meta-analysis, the prevalence of diabetic retinopathy in Latin America and the Caribbean for patients with T1DM was 40.6% and for patients with T2DM it was 37.3%, but the evidence is very uncertain. PDR was twice as high in T1DM than in T2DM, (12.4% vs 5.8%) and NPDR was higher in T2DM than in T1DM (25.0% vs 30.6%). In meta-regression, the mean age in T1DM and duration of diabetes in T2DM explained the heterogeneity of diabetic retinopathy prevalence. The cumulative incidence of retinopathy from one study was 39.6% at 9 years of follow-up. Based on the high heterogeneity, high risk of bias, and publication bias we judged the included evidence to have very low certainty.

### Comparison with other studies

In the present review we provide a better estimate of the prevalence of diabetic retinopathy in Latin America and the Caribbean compared to a global systematic review (22.3%; 95% CI: 19.7 to 25.0), where the number of studies in the region was limited. We found a higher prevalence of the disease, even higher than in Asian countries (28.0%; 95% CI: 24.0 to 33.0) [9]. This difference could be due to the fact that in Latin America and the Caribbean there is a high prevalence of risk factors for the development of diabetic retinopathy such as hypertension, obesity, cardiovascular disease and the intake of foods rich in sugar and fats [71–74], in addition to the lack of implementation of prevention strategies in comparison to more developed countries [75]. It should be taken into account that the certainty of the evidence of the prevalences in T1DM and T2DM is very low and could be overestimated by the number of studies in primary care centers and hospitals and by the bias effect of small populations, so that future studies with better methodological quality could modify our findings.

Only one study reported a cumulative incidence of 39.6% of diabetic retinopathy after 9 years of follow-up, i.e. an annual incidence of approximately 4.4%. Similarly, a global systematic review included 8 studies and found that the cumulative annual incidence ranged from 2.2 to 12.7%, and could not report a pooled incidence because of the heterogeneity of the studies [11].

### Proliferative and non-proliferative diabetic retinopathy

We found that patients with T1DM have twice the prevalence of PDR than patients with T2DM. This is due to the high rates of poor control in patients with T1DM. The patient with T1DM requires greater resources from the health system to achieve these goals. The limitations of the health system in Latin America and the Caribbean do not ensure the adequate treatment of this type of patients. The higher number of NPDR could be due to the fact that the onset of T2DM is not necessarily at debut, but is present previously. Therefore, retinopathy screening is done from the debut unlike T1DM patients who are screened at 5 years. In Asian countries, a prevalence of NPDR of 27% and PDR of 6% in patients with T2DM was reported [9]. This indicates that efforts should be oriented to comply with detection strategies and provide timely treatment [76, 77].

### Diagnostic criteria

Of the 6 diagnostic criteria reported in the studies, we found that there is a lower prevalence of diabetic retinopathy in T2DM when using the L' Esperance diagnostic criterion (6.0%; 95% CI: 2.8 to 11.1). This could be because it is the oldest compared to the other criteria used and it uses different assessments for diagnosis [78], likewise, only one study reported this diagnostic criterion and had an inadequate sample as reported in the risk of bias. On the other hand, only 1 study reported the prevalence with SDRGS (17.3%; 95% CI: 15.2 to 19.5), which showed an equally low prevalence, this could be due to the fact that it was a study with patients attending primary care, being those who attend mostly people with health problems.

### Heterogeneity

We have found high heterogeneity in the overall pooled prevalence and even in the analysis by subgroups. This is a result of clinical and methodological differences in the studies. However, we only included studies with standardized diagnostic criteria and excluded diagnosis by less reliable methods such as self-reporting, medical records, among others. Likewise, variation in prevalence was observed depending on whether the studies were community-based, primary care center-based or hospital-based. It is reported that the prevalence in population-based studies tends to be lower than in hospital-based studies [79]. In addition, meta-regression found an association of retinopathy prevalence with age and duration of diabetes, which is consistent with greater exposure to hyperglycemia and a greater presence of diabetes-associated risk factors with longer duration of disease [80, 81].

### Limitations of the included studies

The included studies have a number of limitations. First, we found prevalence studies from only 10 countries out of 33 countries in Latin America and the Caribbean. Likewise, only 1 study was found that reported the incidence of diabetic retinopathy in the region. Of the 44 studies included, 12 did not specify the type of DM, which made it impossible to include them in the quantitative synthesis. Many of the data were based on hospital series and only 11 studies were based on community data.

## Implications and recommendations

In the present review we found a high prevalence of diabetic retinopathy in patients with T1DM and T2DM, apparently much higher than in other regions of the world, but the evidence is very uncertain. The risk of blindness in patients with DM is estimated to be 25 times higher than in the rest of the population. Without timely treatment, more than 50% will result in total blindness within 5 years [23]. Resulting in significant social and financial consequences [82]. Furthermore, PDR is also associated with an increased risk of cardiovascular death [83]. In a context where metabolic control including blood pressure, lipid profile and glycemia is only 11%. An upward trend in micro and macrovascular complications of diabetes is predicted. In contrast to morbidity in developed countries, where macrovascular complications are on the decline. Therefore, we suggest that the strategies and health policies already implemented for the early detection and prevention of diabetic retinopathy and all its complications in Latin American and Caribbean countries should be complied with or improved. Given the current trend, a high burden of disease is expected, with high costs and poor quality of life for patients.

Future studies on the prevalence and incidence of diabetic retinopathy in the rest of the countries of Latin America and the Caribbean are needed. Above all, studies in the general population, with randomized sampling, sufficient sample sizes, with validated criteria, and independent measurement processes for extrapolation with greater certainty of evidence in the region. It is also necessary that studies report the type of DM being assessed and give details about patients' age, sex, duration of diabetes, and HbA1c. In addition, there is a need for studies that assess differences in prevalences when using different diagnostic criteria in the same population.

## Limitations and strengths

Although this systematic review provides epidemiological data for the health professional on diabetic retinopathy in Latin America and the Caribbean, some limitations should be considered. We did not evaluate how the ophthalmological assessment was performed in each study. The interpretation of the differences in the subgroup analyses was done at the authors' discretion, because there is no test for this purpose. In addition, the meta-regression analyses were post-hoc and should be interpreted with caution, since many of them have little data and future studies could increase the precision of the results. On the other hand, our work has important strengths. We performed an extensive search without restriction of language or date of publication to incorporate all studies from Latin America and the Caribbean. We present the description by subgroups according to T1DM or T2DM. We selected articles that directly measured diabetic retinopathy with validated classifications. We performed multiple techniques to assess heterogeneity. Finally, we assessed the certainty of evidence according to GRADE.

## Conclusion

Two out of five patients with T1DM or T2DM may have diabetic retinopathy in Latin America and the Caribbean, but the evidence is very uncertain (very low certainty of evidence). A relatively high prevalence compared to other regions. The high heterogeneity between and within countries, even describing according to type of DM, can be explained by the type of population and methodological aspects. We suggest that diabetic retinopathy should be considered a major public health problem, and policies and strategies for early detection and opportunely treatment should be proposed. However, prevalence and incidence studies based on general population and with good quality in Latin American and Caribbean countries are still required to achieve an estimate with better external validity and certainty of evidence.

## Supporting information

**S1 Fig. Sensitivity analysis of the variation in prevalence excluding each individual study in T1DM.**
(DOCX)

**S2 Fig. Sensitivity analysis of the variation in prevalence excluding each individual study in T2DM.**
(DOCX)

**S3 Fig. Funnel plot of the overall prevalence of diabetic retinopathy in T1DM.**
(DOCX)

**S4 Fig. Funnel plot of the overall prevalence of diabetic retinopathy in T2DM.**
(DOCX)

**S1 Table. PRISMA 2020 checklist.**
(DOCX)

**S2 Table. Search strategy.**
(DOCX)

**S3 Table. Excluded studies reviewed in full text.**
(DOCX)

**S4 Table. Characteristics of the included study assessing the incidence of diabetic retinopathy in Latin America and the Caribbean (n = 1).**
(DOCX)

**S5 Table. Meta-regression models of diabetic retinopathy in Latin America and the Caribbean countries.**
(DOCX)

**S6 Table. Risk of bias of included studies using the JBI Critical Appraisal Tool for prevalence.**
(DOCX)

## Author Contributions

**Conceptualization:** Sebastian A. Medina-Ramirez, David R. Soriano-Moreno.

**Data curation:** Sebastian A. Medina-Ramirez, David R. Soriano-Moreno, Kimberly G. Tuco, Sharong D. Castro-Diaz.

**Formal analysis:** David R. Soriano-Moreno.

**Investigation:** Sebastian A. Medina-Ramirez, David R. Soriano-Moreno, Kimberly G. Tuco, Sharong D. Castro-Diaz, Josmel Pacheco-Mendoza.

**Methodology:** Sebastian A. Medina-Ramirez, David R. Soriano-Moreno, Josmel Pacheco-Mendoza, Marlon Yovera-Aldana.

**Project administration:** Sebastian A. Medina-Ramirez.

**Software:** David R. Soriano-Moreno.

**Supervision:** Sebastian A. Medina-Ramirez, David R. Soriano-Moreno, Rosa Alvarado-Villacorta, Marlon Yovera-Aldana.

**Writing – original draft:** Sebastian A. Medina-Ramirez, David R. Soriano-Moreno, Kimberly G. Tuco, Sharong D. Castro-Diaz, Rosa Alvarado-Villacorta, Josmel Pacheco-Mendoza, Marlon Yovera-Aldana.

**Writing – review & editing:** Sebastian A. Medina-Ramirez, David R. Soriano-Moreno, Kimberly G. Tuco, Sharong D. Castro-Diaz, Rosa Alvarado-Villacorta, Josmel Pacheco-Mendoza, Marlon Yovera-Aldana.

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
