## [Decision Letter · Decision Letter 0]

22 May 2023

PONE-D-23-08493Prevalence and incidence of diabetic retinopathy in Latin America and the Caribbean: a systematic review and meta-analysisPLOS ONE

Dear Dr. Soriano-Moreno,

Thank you for submitting your manuscript to PLOS ONE. After careful consideration, we feel that it has merit but does not fully meet PLOS ONE’s publication criteria as it currently stands. Therefore, we invite you to submit a revised version of the manuscript that addresses the points raised during the review process.

Congratulations on the manuscript! The text presents findings of great relevance to our understanding of the topic. However, both reviewers have identified important issues that need to be addressed by the authors. The aspects evaluated by the reviewers will undoubtedly enhance the value of the manuscript.

We look forward to receiving your revised manuscript.

Kind regards,

Ricardo de Mattos Russo Rafael, Ph.D.

Academic Editor

PLOS ONE

Journal Requirements:

2. We note that Figure 3 in your submission contain [map/satellite] images which may be copyrighted. All PLOS content is published under the Creative Commons Attribution License (CC BY 4.0), which means that the manuscript, images, and Supporting Information files will be freely available online, and any third party is permitted to access, download, copy, distribute, and use these materials in any way, even commercially, with proper attribution. For these reasons, we cannot publish previously copyrighted maps or satellite images created using proprietary data, such as Google software (Google Maps, Street View, and Earth). For more information, see our copyright guidelines: http://journals.plos.org/plosone/s/licenses-and-copyright.

1. You may seek permission from the original copyright holder of Figure 3 to publish the content specifically under the CC BY 4.0 license.  

Reviewers' comments:

Reviewer's Responses to Questions

**Comments to the Author**

1. Is the manuscript technically sound, and do the data support the conclusions?

Reviewer #1: Partly

Reviewer #2: Yes

2. Has the statistical analysis been performed appropriately and rigorously? 

Reviewer #1: Yes

Reviewer #2: Yes

3. Have the authors made all data underlying the findings in their manuscript fully available?

Reviewer #1: Yes

Reviewer #2: Yes

4. Is the manuscript presented in an intelligible fashion and written in standard English?

Reviewer #1: No

Reviewer #2: Yes

5. Review Comments to the Author

Reviewer #1: Dear authors. Thank you very much for giving me the opportunity to be the first to read the manuscript. The prevalence and incidence of diabetic retinopathy in the Latin American and Caribbean population can help public policy makers to define health policies for the early diagnosis and prevention of this disease as a complication of DM in the region.

This systematic review and meta-analysis brought a lot of relevant evidence that was statistically and descriptively summarized. However, to increase the quality of the manuscript, a set of recommendations is highlighted below.

The title needs to describe the population that was the focus of the analysis; therefore, it is recommended to include the expression population in the title - Prevalence and incidence of diabetic retinopathy in the population of Latin America and Caribbean: a systematic review and meta-analysis.

The purpose of the review has different wording between sections of the manuscript. The abstract states that the objective is “to assess the prevalence and incidence of diabetic retinopathy in Latin America and the Caribbean.” Lines 59-60 of the Introduction read: “Therefore, the objective of this systematic review was to identify the prevalence and incidence of diabetic retinopathy in the population of Latin America and the Caribbean, according to the type of DM.”

The introduction of review studies needs to point to the need to prepare the synthesis of evidence based on the lack of available reviews on the phenomenon of interest.

Lines 55 to 58 of the introduction read, based on two meta-analyses (references 4 and 9), that “There is a wide range in the prevalences of diabetic retinopathy of previous meta-analyses.” The manuscript states that the “heterogeneity could be explained by factors such as the type of population, age group, among others. One of the clinical factors that could explain this is the type of DM due to the different times of onset, treatment, and prognosis [citing references 10–12]”. However, he points out that “The reported meta-analyses do not present this analysis separately.” So, the question is whether the analysis will not be done separately; what is different in the proposed meta-analysis compared to those already available in the literature? What is expected from this systematic review study?

Quantitative and meta-analysis reviews must maintain consistency and coherence to increase the recognition of the synthesized evidence among the scientific community and public policymakers.

It informs in the Method section that the review was carried out following the Preferred Reporting Items for Systematic Reviews and Meta-Analysis (PRISMA) guidelines 2020. This tool consists of a guideline for writing (reporting) the systematic review and meta-analysis that needs to be implemented following a strict method. Review institutes such as Cochrane and JBI have literature with indispensable tools for conducting the review to minimize the risk of errors and biases. The reviewers must choose one and follow the steps recommended.

As for the choice of information sources to access the databases, what was the justification for choosing Web of Science (WoS)/Core Collection, WoS/MEDLINE, WoS/Scielo, Scopus, PubMed, and Embase? By reducing the search in these Core Collection/WoS sources (Medline, Scielo), an apparent selection bias compromises the quality of the synthesized evidence. Meta-analyses, both in the Cochrane Methodology and in the JBI®, it is fundamental that the search strategy is conducted in the broader sources of information, including academic productions, for example, information sources from PubMed, the Virtual Health Library/LILACS, etc.

In data extraction, it needs to explain which system was adopted to include the extracted data and thus generate the statistical meta-analysis. For example, in the Cochrane methodology, the Review Manager is adopted; in the JBI methodology, the JBI summary. The item “Risk of bias” is necessary for selecting studies included in the meta-analysis. The way was written, it looks like it is a separate section of the methodology. The PRISMA checklist indicates that it is necessary to “Specify the methods used to assess the risk of bias in the included studies, including details of the tool(s) used, how many reviewers each study and whether they worked independently, and if applicable, details of automation tools used in the process.

There is a need for greater use of the degree of certainty of the evidence (Table 3) in the presentation of results, discussion, and recommendations in order to suggest public health policies, as one of the main purpose of the systematic reviews and meta-analysis.

Many issues pointed out in these comments need to be methodologically clarified.

Reviewer #2: Manuscript comment PONE-D-23-08493

I consider the systematic review and meta-analysis performed by the authors to be of very good methodological quality. The authors considered the indications of the PRISMA guidelines and other scientific consensus to perform a systematic review of prevalence studies. The manuscript adequately evidences a research question, the inclusion criteria for primary studies, the search strategy for each database, the assessment of risk of bias, the assessment of heterogeneity and reproducibility.

In the following, I offer for the authors' consideration the revision and complement of some ideas:

- Update in the introduction the latest estimates of prevalence and incidence of diabetic retinopathy for the Latin American and Caribbean region.

- What was the definition used in the systematic review to classify patients with type 1 diabetes mellitus and type 2 diabetes mellitus? I believe it is pertinent to clarify the multiple definitions of diabetes used in the selected articles (biochemical diagnosis, clinical history, self-report).

- What was the definition of prevalence used for the selection of the articles, and was an adequate distinction made between prevalence and frequency?

- I consider it pertinent to present the results of the risk of bias assessment in graphs that visualize the percentage of compliance in each of the domains evaluated.

- Why did the authors define a cut-off point of less than 50 participants as an exclusion criterion?

- Why did the authors not consider including the Tau2 test in the heterogeneity assessment? Although the most common tests of heterogeneity are Cochran's Q and the I2 statistic, different results can be found. Cochran's Q test is affected by the number of included studies and with the pressure of the studies, and the I2 statistic is affected by the precision of the included studies. Using all three tests to assess heterogeneity can lead to better conclusions and findings.

- I consider complementing the arguments for the selection of subgroups in the methodology for the evaluation of heterogeneity based on epidemiological and clinical criteria.

- I believe that the authors should be more cautious with the conclusion. By stating that 2 out of 5 patients with diabetes develop retinopathy in Latin America and the Caribbean. Most studies were conducted in the hospital setting and did not calculate a sample size for prevalence studies which could overestimate the results and not be representative.

- I consider it pertinent to complement the results by applying a sensitivity analysis.

- Please update the report in the PROSPERO registry.

Thank you very much

6. PLOS authors have the option to publish the peer review history of their article (what does this mean?). If published, this will include your full peer review and any attached files.

Reviewer #1: **Yes: **Cabral, Ivone

Reviewer #2: **Yes: **Juan Pablo Pérez Bedoya

---

## [Author Response · Author response to Decision Letter 0]

24 Aug 2023

Lima, 30 July 2023.

Dear Mr. Editor,

Receive greetings from the authors. We highly appreciate the suggestions provided to enhance the study.

We have carefully considered the comments offered by the reviewers and, accordingly, we have made substantial changes to the manuscript, as indicated below. For these modifications, we used the "track changes" option directly in the main document.

Sincerely,

David R. Soriano-Moreno

 

Editor comments

Reply: Thanks for the suggestion, we have adapted the manuscript according to style requirements.

2. We note that Figure 3 in your submission contain [map/satellite] images which may be copyrighted. All PLOS content is published under the Creative Commons Attribution License (CC BY 4.0), which means that the manuscript, images, and Supporting Information files will be freely available online, and any third party is permitted to access, download, copy, distribute, and use these materials in any way, even commercially, with proper attribution. For these reasons, we cannot publish previously copyrighted maps or satellite images created using proprietary data, such as Google software (Google Maps, Street View, and Earth). For more information, see our copyright guidelines: http://journals.plos.org/plosone/s/licenses-and-copyright.

1. You may seek permission from the original copyright holder of Figure 3 to publish the content specifically under the CC BY 4.0 license. cc

Reply: Thank you for the observation. We contacted the authors of the figure and awaited their response for several weeks; however, we did not get a response. Therefore, we decided to remove the figure from the manuscript.

Reviewer #1

Dear authors. Thank you very much for giving me the opportunity to be the first to read the manuscript. The prevalence and incidence of diabetic retinopathy in the Latin American and Caribbean population can help public policy makers to define health policies for the early diagnosis and prevention of this disease as a complication of DM in the region. This systematic review and meta-analysis brought a lot of relevant evidence that was statistically and descriptively summarized. However, to increase the quality of the manuscript, a set of recommendations is highlighted below.

1. The title needs to describe the population that was the focus of the analysis; therefore, it is recommended to include the expression population in the title - Prevalence and incidence of diabetic retinopathy in the population of Latin America and Caribbean: a systematic review and meta-analysis.

Reply: We accepted the suggestion; the title was changed “Prevalence and incidence of diabetic retinopathy in patients with diabetes of Latin America and the Caribbean: a systematic review and meta-analysis”.

2. The purpose of the review has different wording between sections of the manuscript. The abstract states that the objective is “to assess the prevalence and incidence of diabetic retinopathy in Latin America and the Caribbean.” Lines 59-60 of the Introduction read: “Therefore, the objective of this systematic review was to identify the prevalence and incidence of diabetic retinopathy in the population of Latin America and the Caribbean, according to the type of DM.”

Reply: Thanks for the comment, we standardized the wording of the objective: “This systematic review aimed to assess the prevalence and incidence of diabetic retinopathy in patients with diabetes of Latin America and the Caribbean.”

3. The introduction of review studies needs to point to the need to prepare the synthesis of evidence based on the lack of available reviews on the phenomenon of interest.

Reply: We appreciate the comment, we added the following in the introduction: “A systematic review with meta-analysis was carried out at a global level and found a prevalence of 22.3% [4]. Another review conducted in Asia reported a prevalence of 28% [9]. Likewise, a systematic review reported a prevalence in Africa that varied between 30.2% and 31.6% [10]. Additionally, a systematic review evaluated the annual incidence of diabetic retinopathy in Asia, North America, the Caribbean and Africa varied from 2.2% to 12.7% [11]. However, in our population there is a knowledge gap about the prevalence and incidence of diabetic retinopathy.”

4. Lines 55 to 58 of the introduction read, based on two meta-analyses (references 4 and 9), that “There is a wide range in the prevalences of diabetic retinopathy of previous meta-analyses.” The manuscript states that the “heterogeneity could be explained by factors such as the type of population, age group, among others. One of the clinical factors that could explain this is the type of DM due to the different times of onset, treatment, and prognosis [citing references 10–12]”. However, he points out that “The reported meta-analyses do not present this analysis separately.” So, the question is whether the analysis will not be done separately; what is different in the proposed meta-analysis compared to those already available in the literature? What is expected from this systematic review study? Quantitative and meta-analysis reviews must maintain consistency and coherence to increase the recognition of the synthesized evidence among the scientific community and public policymakers.

Reply: Thanks for the comment. Regarding the differences between our proposed meta-analysis and existing reviews in the literature, we would like to highlight that our research is distinguished by its specific focus on differentiating the prevalence and incidence of diabetic retinopathy according to the type of diabetes in the population of Latin America and the Caribbean. This differentiation is essential, since the type of diabetes may have different age of onset, treatments and prognoses, which influence the prevalence of diabetic retinopathy. We added the following in the introduction “However, in our population there is a knowledge gap about the prevalence and incidence of diabetic retinopathy. This is relevant since the prevalence can be influenced by various factors, such as the type of population evaluated, the type of DM, the time of onset of the disease, the different treatments applied, the age of the patients, among others [12–14]. In addition, it is important to highlight that existing reviews do not usually evaluate the prevalence and incidence of diabetic retinopathy according to the type of DM.”

5. It informs in the Method section that the review was carried out following the Preferred Reporting Items for Systematic Reviews and Meta-Analysis (PRISMA) guidelines 2020. This tool consists of a guideline for writing (reporting) the systematic review and meta-analysis that needs to be implemented following a strict method. Review institutes such as Cochrane and JBI have literature with indispensable tools for conducting the review to minimize the risk of errors and biases. The reviewers must choose one and follow the steps recommended.

Reply: Thanks for the comment, in our research paper, we have adhered to the PRISMA guideline for the overall structure of the article. We have predominantly followed the methodology proposed by JBI for the majority of the steps, including search, selection, extraction, and assessment of risk of bias. Therefore, in the methods section, we have provided a detailed description of the JBI methodology, emphasizing its application in our study: “We performed a systematic review following the methodology proposed by Joanna Briggs Institute Manual for Evidence Synthesis: Systematic reviews of prevalence and incidence [15] and the Preferred Reporting Items for Systematic Reviews and Meta-Analysis (PRISMA) guidelines 2020 [16] S1 table.” 

6. As for the choice of information sources to access the databases, what was the justification for choosing Web of Science (WoS)/Core Collection, WoS/MEDLINE, WoS/Scielo, Scopus, PubMed, and Embase? By reducing the search in these Core Collection/WoS sources (Medline, Scielo), an apparent selection bias compromises the quality of the synthesized evidence. Meta-analyses, both in the Cochrane Methodology and in the JBI®, it is fundamental that the search strategy is conducted in the broader sources of information, including academic productions, for example, information sources from PubMed, the Virtual Health Library/LILACS, etc.

Reply: Thanks for the comment. Regarding the choice of information sources to access the databases, the justification for selecting Web of Science (WoS)/Core Collection, WoS/MEDLINE, WoS/Scielo, Scopus, PubMed, and Embase was based on the following factors:

• WoS/Core Collection, WoS/MEDLINE, WoS/Scielo, Scopus, PubMed, and Embase are well-established and widely recognized databases, known for their extensive coverage of academic literature across various disciplines, including medicine and healthcare.

• Cochrane Methodology and JBI recommend using Embase and Medline (from WoS) as two of the essential databases for conducting systematic reviews and meta-analyses due to their comprehensive coverage of medical literature.

• Scopus and WoS are considered to be the most important structured databases globally, containing a vast amount of information from diverse sources, thereby providing a broader perspective on the available evidence.

• Scielo (from WoS) was chosen because it includes research from Latin American countries, offering valuable information from the region that may not be as widely covered in other databases.

• PubMed, a free specialized database in medicine, was included to complement the information found in MEDLINE and enhance the overall comprehensiveness of the search.

By using this combination of databases, the aim is to cover a wide range of information sources, ensuring that both quantity and quality of the synthesized evidence are not compromised. However, it is worth noting that LILACS, another database suggested in the commentary, was not included due to its poor export management, which requires manual selection of individual articles, making it less practical. For this reason, at the beginning of the systematic review, the authors decided not to include it.

7. In data extraction, it needs to explain which system was adopted to include the extracted data and thus generate the statistical meta-analysis. For example, in the Cochrane methodology, the Review Manager is adopted; in the JBI methodology, the JBI summary. The item “Risk of bias” is necessary for selecting studies included in the meta-analysis. The way was written, it looks like it is a separate section of the methodology. The PRISMA checklist indicates that it is necessary to “Specify the methods used to assess the risk of bias in the included studies, including details of the tool(s) used, how many reviewers each study and whether they worked independently, and if applicable, details of automation tools used in the process.

Reply: The data to be extracted were selected according to the objectives defined in the protocol. The information was extracted in an Excel spreadsheet. The Excel information was imported into the statistical program (STATA) where the analysis was performed. The risk of bias is an important item to evaluate whether this factor influences the heterogeneity of the results. However, we did not decide to use this criterion as an inclusion criterion of the studies for the meta-analysis. We added the details of the tool used. We changed the term "Three authors" to "The authors".

8. There is a need for greater use of the degree of certainty of the evidence (Table 3) in the presentation of results, discussion, and recommendations in order to suggest public health policies, as one of the main purpose of the systematic reviews and meta-analysis. Many issues pointed out in these comments need to be methodologically clarified.

Reply: Thank you for the comment, we have rephrased the results considering certainty. We have used the GRADE 26 article detailing how to phrase the results according to the certainty of the evidence (https://www.jclinepi.com/article/S0895-4356(19)30416-0/fulltext). We have quoted this article in methods: "We communicated the findings of the main results using the informative statements proposed by GRADE". We have changed the phrasing as follows: 

Abstract: "The overall prevalence of retinopathy in patients with T1DM was 40.6% (95% CI: 34.7 to 46.6; I2: 92.1%) and in T2DM was 37.3% (95% CI: 31.0 to 43.8; I2: 97.7), but the evidence is very uncertain (very low certainty of evidence)" and "Two out of five patients with T1DM or T2DM may present diabetic retinopathy in Latin America and the Caribbean, but the evidence is very uncertain".

Results: "In patients with T1DM, the prevalence of diabetic retinopathy was 40.6% (95% CI: 34.7 to 46.6; I2: 92.1%), but the evidence is very uncertain. For patients with T2DM, the prevalence was 37.3% (95% CI: 31.0 to 43.8; I2: 97.7%), but the evidence is very uncertain (Table 2 and Fig 2)."

Discussion: "In this systematic review and meta-analysis, the prevalence of diabetic retinopathy in Latin America and the Caribbean for patients with T1DM was 40.6% and for patients with T2DM was 37.3%, but the evidence is very uncertain."

Recommendations: "In the present review we found a high prevalence of diabetic retinopathy in patients with T1DM and T2DM, apparently much higher than in other regions of the world, but the evidence is very uncertain."

Conclusion: "Two out of five patients with T1DM or T2DM may have diabetic retinopathy in Latin America and the Caribbean, but the evidence is very uncertain (very low certainty of evidence)."

Reviewer #2:

I consider the systematic review and meta-analysis performed by the authors to be of very good methodological quality. The authors considered the indications of the PRISMA guidelines and other scientific consensus to perform a systematic review of prevalence studies. The manuscript adequately evidences a research question, the inclusion criteria for primary studies, the search strategy for each database, the assessment of risk of bias, the assessment of heterogeneity and reproducibility. In the following, I offer for the authors' consideration the revision and complement of some ideas:

1. Update in the introduction the latest estimates of prevalence and incidence of diabetic retinopathy for the Latin American and Caribbean region.

Reply: We appreciate the suggestion. However, we would like to highlight that to our knowledge there is currently a gap in the evidence regarding other prevalence and incidence estimates specifically for this region. We added the following in the introduction: “A systematic review with meta-analysis was carried out at a global level and found a prevalence of 22.3% [4]. Another review conducted in Asia reported a prevalence of 28% [9]. Likewise, a systematic review reported a prevalence in Africa that varied between 30.2% and 31.6% [10]. Additionally, a systematic review evaluated the annual incidence of diabetic retinopathy in Asia, North America, the Caribbean and Africa varied from 2.2% to 12.7% [11]. However, in our population there is a knowledge gap about the prevalence and incidence of diabetic retinopathy. This is relevant since the prevalence can be influenced by various factors, such as the type of population evaluated, the type of DM, the time of onset of the disease, the different treatments applied, the age of the patients, among others [12–14]. In addition, it is important to highlight that existing reviews do not usually evaluate the prevalence and incidence of diabetic retinopathy according to the type of DM.”

2. What was the definition used in the systematic review to classify patients with type 1 diabetes mellitus and type 2 diabetes mellitus? I believe it is pertinent to clarify the multiple definitions of diabetes used in the selected articles (biochemical diagnosis, clinical history, self-report).

Reply: Thanks for the comment, we added the following in eligibility criteria: “The definition of DM was taken from what was reported by the study, in case the study did not explicitly mention the type of diabetes, it was considered as T1DM and T2DM”.

3. What was the definition of prevalence used for the selection of the articles, and was an adequate distinction made between prevalence and frequency?

Reply: We appreciate your feedback, we added the following in Statistical analyses: “The prevalence of retinopathy according to type of DM was calculated using the number of patients with diabetes as the denominator and the number of cases of diabetic retinopathy as the numerator, enabling us to calculate the prevalence”.

4. I consider it pertinent to present the results of the risk of bias assessment in graphs that visualize the percentage of compliance in each of the domains evaluated.

Reply: Thank you for your suggestion. We present the results of the risk of bias assessment in Fig 4 that represents the percentage of compliance for each evaluated domain.

5. Why did the authors define a cut-off point of less than 50 participants as an exclusion criterion?

Reply: We appreciate your comments and would like to address the concern you raised regarding the inclusion of research studies with a population size of fewer than 50 participants. This choice was by our awareness of the potential limitations associated with smaller sample sizes. In preliminary searches we observed such studies often generated heterogeneous information, which could affect the reliability and generalizability of the findings. We added the following in methods: “We opted to exclude studies with a sample size of fewer than 50 patients due to the potential impact on the reliability and generalizability of the findings.”

6. Why did the authors not consider including the Tau2 test in the heterogeneity assessment? Although the most common tests of heterogeneity are Cochran's Q and the I2 statistic, different results can be found. Cochran's Q test is affected by the number of included studies and with the pressure of the studies, and the I2 statistic is affected by the precision of the included studies. Using all three tests to assess heterogeneity can lead to better conclusions and findings.

Reply: Thanks for the comment. Although Tau2 is insensitive to the number of studies and precision, we believe it would not change the final interpretation of heterogeneity, as the I2 takes into account the magnitude, direction, and strength of heterogeneity.

7. I consider complementing the arguments for the selection of subgroups in the methodology for the evaluation of heterogeneity based on epidemiological and clinical criteria.

Reply: Thank you for the suggestion. Certainly, incorporating epidemiological and clinical criteria when selecting subgroups in assessing heterogeneity improves the understanding of the studied population and identifies effect modifiers. We have cited the following articles in the statistical analysis section:

● Yau JW, Rogers SL, Kawasaki R, Lamoureux EL, Kowalski JW, Bek T, Chen SJ, Dekker JM, Fletcher A, Grauslund J, Haffner S, Hamman RF, Ikram MK, Kayama T, Klein BE, Klein R, Krishnaiah S, Mayurasakorn K, O'Hare JP, Orchard TJ, Porta M, Rema M, Roy MS, Sharma T, Shaw J, Taylor H, Tielsch JM, Varma R, Wang JJ, Wang N, West S, Xu L, Yasuda M, Zhang X, Mitchell P, Wong TY; Meta-Analysis for Eye Disease (META-EYE) Study Group. Global prevalence and major risk factors of diabetic retinopathy. Diabetes Care. 2012 Mar;35(3):556-64. doi: 10.2337/dc11-1909. Epub 2012 Feb 1. PMID: 22301125; PMCID: PMC3322721.

● Roto A, Farah R, Al-Imam M, Q Al-Sabbagh M, Abu-Yaghi N. Prevalence, characteristics and risk factors of diabetic retinopathy in type 2 diabetes mellitus patients in Jordan: a cross-sectional study. J Int Med Res. 2022 Aug;50(8):3000605221115156. doi: 10.1177/03000605221115156. PMID: 35938493; PMCID: PMC9364199.

● Teo ZL, Tham YC, Yu M, Chee ML, Rim TH, Cheung N, Bikbov MM, Wang YX, Tang Y, Lu Y, Wong IY, Ting DSW, Tan GSW, Jonas JB, Sabanayagam C, Wong TY, Cheng CY. Global Prevalence of Diabetic Retinopathy and Projection of Burden through 2045: Systematic Review and Meta-analysis. Ophthalmology. 2021 Nov;128(11):1580-1591. doi: 10.1016/j.ophtha.2021.04.027. Epub 2021 May 1. PMID: 33940045.

8. I believe that the authors should be more cautious with the conclusion. By stating that 2 out of 5 patients with diabetes develop retinopathy in Latin America and the Caribbean. Most studies were conducted in the hospital setting and did not calculate a sample size for prevalence studies which could overestimate the results and not be representative.

Reply: We agree with your comment. We have changed the phrasing of the conclusion based on the GRADE 26 article (https://www.jclinepi.com/article/S0895-4356(19)30416-0/fulltext). Conclusion: "Two out of five patients with T1DM or T2DM may have diabetic retinopathy in Latin America and the Caribbean, but the evidence is very uncertain (very low certainty of evidence)."

9. I consider it pertinent to complement the results by applying a sensitivity analysis.

Reply: We accepted the suggestion. We added sensitivity analyses in supplementary materials S5 fig. and S6 fig. In the results we wrote the following: “When excluding each individual study, the pooled prevalence of diabetic retinopathy for T1DM ranged from 40.3% (95% CI: 34.2 to 46.4) to 43.0% (95% CI: 36.6 to 49.5) (S5 fig.) and for T2DM ranged from 34.4% (95% CI: 28.4 to 40.6) to 38.4% (95% CI: 31.8 to 45.0) (S6 fig.)”.

10. Please update the report in the PROSPERO registry.

Reply: Thanks for the suggestion, we have updated the report in PROSPERO.

---

## [Decision Letter · Decision Letter 1]

16 Oct 2023

PONE-D-23-08493R1Prevalence and incidence of diabetic retinopathy in patients with diabetes of Latin America and the Caribbean: a systematic review and meta-analysisPLOS ONE

Dear Dr. Soriano-Moreno,

Thank you for submitting your manuscript to PLOS ONE. After careful consideration, we feel that it has merit but does not fully meet PLOS ONE’s publication criteria as it currently stands. Therefore, we invite you to submit a revised version of the manuscript that addresses the points raised during the review process.

The reviewers have raised significant methodological issues that need to be elaborated upon by the authors. Consequently, I kindly request a careful consideration of these aspects.

We look forward to receiving your revised manuscript.

Kind regards,

Ricardo de Mattos Russo Rafael, Ph.D.

Academic Editor

PLOS ONE

Journal Requirements:

Reviewers' comments:

Reviewer's Responses to Questions

**Comments to the Author**

1. If the authors have adequately addressed your comments raised in a previous round of review and you feel that this manuscript is now acceptable for publication, you may indicate that here to bypass the “Comments to the Author” section, enter your conflict of interest statement in the “Confidential to Editor” section, and submit your "Accept" recommendation.

Reviewer #1: All comments have been addressed

Reviewer #2: All comments have been addressed

2. Is the manuscript technically sound, and do the data support the conclusions?

Reviewer #1: Yes

Reviewer #2: Yes

3. Has the statistical analysis been performed appropriately and rigorously? 

Reviewer #1: Yes

Reviewer #2: Yes

4. Have the authors made all data underlying the findings in their manuscript fully available?

Reviewer #1: Yes

Reviewer #2: Yes

5. Is the manuscript presented in an intelligible fashion and written in standard English?

Reviewer #1: Yes

Reviewer #2: Yes

6. Review Comments to the Author

Reviewer #1: Thanks for the opportunity to review the manuscript again. We can find that many improvements were registered. The authors appropriately reviewed and addressed the recommendations in a new version of the manuscript. However, some relevant methodological issues need to be addressed to avoid the risk of bias.

In order to have consistency, the search strategy must be written according to the JBI methodology for systematic review. There seems to be little understanding of the difference between an information source and a database. So far, it needs to be more clarity in the description of the search strategy and not in how the search was conducted. The text conveys that only journals in the indexing based on the Web of Science were accessed. See at the topic "Literature search and study selection": "A systematic search was performed in six databases: Web of Science (WoS)/Core Collection, WoS/MEDLINE, WoS/Scielo, Scopus, PubMed and Embase. It would be an inconsistency since texts duplicated in multiple databases were excluded. See the JBI Manual, the Systematic Review chapter, for a better presentation of the sources accessed.

Throughout the method section's text, the JBI's name is recorded in its previous form. Please check the link (https://jbi.global/news/article/revealed-new-jbi) for the current name and the justification for its suitability. JBI's new name and logo signify its evolution and capture how the organization is known internationally today. That is, as 'JBI'—for example, JBI critical appraisal tools and JBI methodology.

Reviewer #2: (No Response)

7. PLOS authors have the option to publish the peer review history of their article (what does this mean?). If published, this will include your full peer review and any attached files.

Reviewer #1: **Yes: **Ivone Evangelista Cabral. Faculdade de Enfermagem. Universidade do Estado do Rio de Janeiro. Core Staff Member of The Brazilian Centre for Evidence-based Health Care: A JBI Centre of Excellence (JBI Brazil)

Reviewer #2: **Yes: **Juan Pablo Pérez Bedoya

---

## [Author Response · Author response to Decision Letter 1]

11 Nov 2023

Response to the reviewers

Reviewer comments

Comment: Thanks for the opportunity to review the manuscript again. We can find that many improvements were registered. The authors appropriately reviewed and addressed the recommendations in a new version of the manuscript. However, some relevant methodological issues need to be addressed to avoid the risk of bias.

In order to have consistency, the search strategy must be written according to the JBI methodology for systematic review. There seems to be little understanding of the difference between an information source and a database. So far, it needs to be more clarity in the description of the search strategy and not in how the search was conducted. The text conveys that only journals in the indexing based on the Web of Science were accessed. See at the topic "Literature search and study selection": "A systematic search was performed in six databases: Web of Science (WoS)/Core Collection, WoS/MEDLINE, WoS/Scielo, Scopus, PubMed and Embase. It would be an inconsistency since texts duplicated in multiple databases were excluded. See the JBI Manual, the Systematic Review chapter, for a better presentation of the sources accessed.

Reply: Thank you very much for your comment, we have checked the JBI manual, chapter 5: systematic reviews of prevalence and incidence, section 5.5.5 search strategy. For clarification, we have changed the paragraph: “A systematic search was performed in six information sources: Web of Science (WoS)/Core Collection, WoS/MEDLINE, WoS/Scielo, Scopus, PubMed and Embase until January 16, 2023.” The search strategy is detailed in S2 table.

Comment: Throughout the method section's text, the JBI's name is recorded in its previous form. Please check the link (https://jbi.global/news/article/revealed-new-jbi) for the current name and the justification for its suitability. JBI's new name and logo signify its evolution and capture how the organization is known internationally today. That is, as 'JBI'—for example, JBI critical appraisal tools and JBI methodology.

Reply: Thanks for the comment, we have changed to “JBI” throughout the manuscript.

---

## [Decision Letter · Decision Letter 2]

4 Dec 2023

PONE-D-23-08493R2Prevalence and incidence of diabetic retinopathy in patients with diabetes of Latin America and the Caribbean: a systematic review and meta-analysisPLOS ONE

Dear Dr. Soriano-Moreno,

Thank you for submitting your manuscript to PLOS ONE. After careful consideration, we feel that it has merit but does not fully meet PLOS ONE’s publication criteria as it currently stands. Therefore, we invite you to submit a revised version of the manuscript that addresses the points raised during the review process.

We are approaching the third round of evaluation, and it is of paramount importance that due consideration be accorded to all the points delineated by the specialized evaluator. It is noteworthy to underscore that one of the evaluators has previously granted approval to the manuscript in prior assessments; however, there remain outstanding matters that demand attention. I wish to apprise the stakeholders that I have also undertaken a comprehensive reevaluation of the manuscript and concur fully with the aspects identified by the current specialized evaluator. We await with eager anticipation the swift implementation of the requisite improvements.

We look forward to receiving your revised manuscript.

Kind regards,

Ricardo de Mattos Russo Rafael, Ph.D.

Academic Editor

PLOS ONE

Journal Requirements:

Reviewers' comments:

Reviewer's Responses to Questions

**Comments to the Author**

1. If the authors have adequately addressed your comments raised in a previous round of review and you feel that this manuscript is now acceptable for publication, you may indicate that here to bypass the “Comments to the Author” section, enter your conflict of interest statement in the “Confidential to Editor” section, and submit your "Accept" recommendation.

Reviewer #1: All comments have been addressed

2. Is the manuscript technically sound, and do the data support the conclusions?

Reviewer #1: Yes

3. Has the statistical analysis been performed appropriately and rigorously? 

Reviewer #1: Yes

4. Have the authors made all data underlying the findings in their manuscript fully available?

Reviewer #1: Yes

5. Is the manuscript presented in an intelligible fashion and written in standard English?

Reviewer #1: Yes

6. Review Comments to the Author

Reviewer #1: After a new reading, only one inconsistency in the writing was identified: a methodological mistake for systematic reviews according to the JBI methodology.

The response to the peer review recommendation states that “A systematic search was performed in six information sources: Web of Science (WoS)/Core Collection, WoS/MEDLINE, WoS/Scielo, Scopus, PubMed and Embase until January 16, 2023. ”

The Core Collection is a source of information with Web of Science as one of the journal indexing bases. Science Direct is the source of information, with Scopus as one of the journal indexing bases. Medline is a journal indexing database that can be accessed through the PubMed Library. It is not possible to access journals indexed in MedLine through WoS.

The same information needs to be reviewed concerning collecting journals indexed in SciELO.

In this sense, the information in the manuscript does not correspond to what PONE readers can use in the future. However, when conducting a review whose starting point is to include only journals accessed by the Core Collection source of information or journals from the WoS index database, it produces a bias in the analysis of evidence.

7. PLOS authors have the option to publish the peer review history of their article (what does this mean?). If published, this will include your full peer review and any attached files.

Reviewer #1: **Yes: **Ivone Evangelista Cabral

Associate Professor. College of Nursing. State University of Rio de Janeiro.

Core Staff Member of JBI Brazil

---

## [Author Response · Author response to Decision Letter 2]

12 Dec 2023

Reviewer comments

Comment: After a new reading, only one inconsistency in the writing was identified: a methodological mistake for systematic reviews according to the JBI methodology.

The response to the peer review recommendation states that “A systematic search was performed in six information sources: Web of Science (WoS)/Core Collection, WoS/MEDLINE, WoS/Scielo, Scopus, PubMed and Embase until January 16, 2023. ”

The Core Collection is a source of information with Web of Science as one of the journal indexing bases. Science Direct is the source of information, with Scopus as one of the journal indexing bases. Medline is a journal indexing database that can be accessed through the PubMed Library. It is not possible to access journals indexed in MedLine through WoS.

The same information needs to be reviewed concerning collecting journals indexed in SciELO.

In this sense, the information in the manuscript does not correspond to what PONE readers can use in the future. However, when conducting a review whose starting point is to include only journals accessed by the Core Collection source of information or journals from the WoS index database, it produces a bias in the analysis of evidence.

Reply: Thank you for your valuable feedback. In response to your suggestion, we have opted to revise the term "sources of information" to simply "sources," aligning with Cochrane's general usage in describing the search scope of systematic reviews. Regarding Web of Science, it is important to note that it comprises a collection of databases. Although a general search is possible across all databases included in Web of Science without specifying a particular database, we recognize the importance of adapting the search strategy for each database to ensure precision. For instance, when searching MEDLINE through Web of Science, MeSH terms are employed, whereas Core Collection necessitates a different approach without the use of MeSH terms. Therefore, the search strategies for Core Collection, MEDLINE and Scielo are different. Regarding MEDLINE in Web of Science, it is available from 1950 to the present. Once again, we appreciate your thorough review and constructive input, which will significantly enhance the accuracy and transparency of our systematic review.

---

## [Editor Report · Decision Letter 3]

27 Dec 2023

Prevalence and incidence of diabetic retinopathy in patients with diabetes of Latin America and the Caribbean: a systematic review and meta-analysis

PONE-D-23-08493R3

Dear Dr. Soriano-Moreno,

We’re pleased to inform you that your manuscript has been judged scientifically suitable for publication and will be formally accepted for publication once it meets all outstanding technical requirements.

Kind regards,

Ricardo de Mattos Russo Rafael, Ph.D.

Academic Editor

PLOS ONE

---

## [Editor Report · Acceptance letter]

11 Jan 2024

PONE-D-23-08493R3 

PLOS ONE

Dear Dr. Soriano-Moreno, 

I'm pleased to inform you that your manuscript has been deemed suitable for publication in PLOS ONE. Congratulations! Your manuscript is now being handed over to our production team.

Kind regards, 

on behalf of

Dr. Ricardo de Mattos Russo Rafael 

Academic Editor

PLOS ONE